# Contrastive Learning from Synthetic Audio Doppelgängers

**Manuel Cherep**[*]
Massachusetts Institute of Technology
mcherep@mit.edu

**Nikhil Singh**[*]
Dartmouth College
nikhil.u.singh@dartmouth.edu

doppelgangers.media.mit.edu

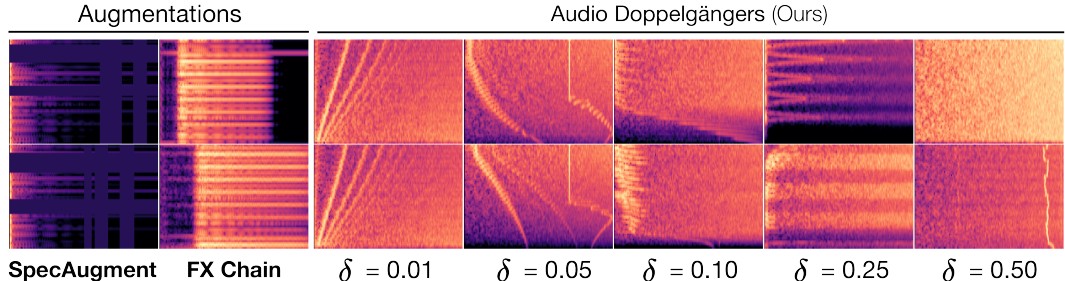

Figure 1: **(Left)** Standard data augmentation techniques for contrastive learning applied to audio spectrograms **(Right)** *Audio Doppelgängers*, our approach synthesizing sounds that are controllably different using perturbed synthesis parameters, shown for different factors $\delta$. These sounds can vary in causally controllable ways beyond what data augmentations can achieve.

## ABSTRACT

Learning robust audio representations currently demands extensive datasets of real-world sound recordings. By applying artificial transformations to these recordings, models can learn to recognize similarities despite subtle variations through techniques like contrastive learning. However, these transformations are only approximations of the true diversity found in real-world sounds, which are generated by complex interactions of physical processes, from vocal cord vibrations to the resonance of musical instruments. We propose a solution to both the data scale and transformation limitations, leveraging synthetic audio. By randomly perturbing the parameters of a sound synthesizer, we generate *audio doppelgängers*—synthetic positive pairs with causally manipulated variations in timbre, pitch, and temporal envelopes. These variations, difficult to achieve through augmentations of existing audio, provide a rich source of contrastive information. Despite the shift to randomly generated synthetic data, our method produces strong representations, outperforming real data on several standard audio classification tasks. Notably, our approach is lightweight, requires no data storage, and has only a single hyperparameter, which we extensively analyze. We offer this method as a complement to existing strategies for contrastive learning in audio, using synthesized sounds to reduce the data burden on practitioners.

## 1 INTRODUCTION

> *"Noises have generally been thought of as indistinct, but this is not true."*
> — **Pierre Schaeffer**, *1986*

The success of modern machine learning algorithms for tasks like audio understanding often hinges on both the quality and quantity of available data. Self-supervised learning methods, like contrastive

---

[*]Equal contribution.

learning, have even been able to leverage unlabeled data, enabling more human-like learning from patterns without needing explicit supervision. However, human perceptual processing is remarkably robust beyond this: for example, the human auditory system can easily recognize sounds across a wide range of variations, such as changes in pitch, timbre, or background noise. Moreover, humans can quickly learn to recognize novel sounds that they encounter in their environment. Replicating this ability to learn from a diverse array of sounds—or "noises," as we might call them—could significantly enhance the efficiency, scalability, and adaptability of machine learning models.

Contrastive learning, which operates by recognizing similarities in the data among negative distractors, often relies on augmentations: transformations of input data that preserve content semantics. This method has been influential in audio representation learning, with specific implementations ranging from spectral masking to temporal jitter to cropping and other methods (Huang et al., 2022; Saeed et al., 2021; Spijkervet & Burgoyne, 2021; Wang & Oord, 2021; Al-Tahan & Mohsenzadeh, 2021; Niizumi et al., 2021; Manocha et al., 2021). Data augmentations, though demonstrably useful, operate at the level of the observed data, not the underlying data-generating process as would be observed in real-world variation. They statistically alter data without directly manipulating the causal mechanisms that produced it, resulting in high correlation between augmented samples, as well as limited control and interpretability.

In our work, we propose a different strategy: using a synthesizer to overcome this barrier, in addition to providing the scalability required for modern pretraining regimes through virtually unlimited data synthesis. A synthesizer can be understood as a system where parameters (relating to psychophysical attributes like pitch, timbre, and loudness) causally influence the generated sound. Modifying these parameters allows us to intervene in the data-generating process in a controllable way to generate positive pairs that vary in terms of their underlying synthesis parameters. Unlike traditional data augmentation techniques, our method generates entirely synthetic audio data from scratch. This approach allows us to control the underlying data-generating process directly, offering a perspective distinct from augmentation of real data.

We formulate an approach in which we randomly synthesize sounds, and then slightly perturb their parameters to generate positive pairs. We call these *audio doppelgängers* (examples in Figure 1); they share a resemblance but are in fact distinct enough to learn from the variation between them. In a way, this approach uses an artificial data source effectively consisting of random synthetic noises but more "natural" differences akin to variation in similar sounds; as Pierre Schaeffer put it, noises are not indistinct. Through a comprehensive set of experiments, we show that models trained this way can yield strong performance on a wide range of downstream tasks, competitive with real audio.

Overall, this work contributes:

1. An approach to synthesizing paired audio examples with a continuously controllable degree of dissimilarity, specified by a simple and interpretable hyperparameter $\delta$.

2. The first study, to our knowledge, of synthetic data methods for audio representation learning.

3. Comprehensive experiments in which we train and compare over 20 model variants across 8 downstream tasks to provide evidence that training with our approach can yield strong results on a wide range of audio processing tasks.

4. An analysis of how these synthetic datasets differ from realistic audio datasets in terms of their auditory features, and how this might contribute to learning effective representations.

## 2 RELATED WORK

### 2.1 LEARNING FROM SYNTHETIC DATA

Synthetic data, artificially generated information rather than collected from real-world sources, has emerged as a valuable tool for learning across various domains (Liu et al., 2023; Silver et al., 2017; Kumar et al., 2020; Meng et al., 2022). By addressing data scarcity, privacy concerns (Tucker et al., 2020; DuMont Schütte et al., 2021), or removing biases (Tan et al., 2020; Ramaswamy et al., 2021), synthetic data offers a promising avenue to complement scarce Longpre et al. (2024) real-world data and further drive progress in machine learning research.

Audio presents unique challenges due to the complexity of waveforms and temporal dependencies. Synthetic data has found applications in subareas like speech recognition (Rosenberg et al., 2019; Rossenbach et al., 2020; Laptev et al., 2020; Fazel et al., 2021; Hu et al., 2022; Gao et al.) leveraging text-to-speech systems for detecting unspoken punctuation (Soboleva et al., 2021), recognizing low-resource languages (Bartelds et al., 2023), increasing acoustic diversity (Chen et al., 2020b) or detecting out-of-vocabulary words (Zheng et al., 2021). However, non-speech audio domains can be highly diverse, requiring more complex approaches to data synthesis. In this domain, synthetic data has been used for specific tasks like timbre-text alignment (Jonason & Sturm, 2022) and vocoding (Wang et al., 2023). The partially synthetic NSynth (Engel et al., 2017) dataset has also been used for pitch estimation and instrument classification. In our work, we tackle the general audio domain, proposing a synthetic data approach that can produce diverse sounds for general-purpose audio representation learning.

In computer vision, synthetic data is more popular and has been employed in different tasks to improve performance (Chen et al., 2019; Ros et al., 2016; Varol et al., 2017; Ionescu et al., 2013; Shakhnarovich et al., 2003; Mayer et al., 2016; Dosovitskiy et al., 2015; Ren & Lee, 2018). While initially focused on using graphics engines to generate photorealistic scenes, recent work has investigated sampling synthetic data from deep generative models (Besnier et al., 2020; Ravuri & Vinyals, 2019; Jahanian et al., 2021; Zhang et al., 2021; Tritrong et al., 2021; Li et al., 2021; Shrivastava et al., 2017; Hoffman et al., 2018; Tian et al., 2024; 2023; Trabucco et al., 2023; Yang et al., 2021; Jeong & Shin, 2021). However, these models aim to produce realistic images and still depend on real image datasets for training or synthesis. Thus, recent work has pushed away from realism, generating synthetic data such as fractals (Kataoka et al., 2020), or through other procedural noise models (Baradad Jurjo et al., 2021; Baradad et al., 2022) to use as training data for visual representation learners. In our work, we also abandon realism and leverage randomly generated synthetic sounds to learn audio representations for downstream tasks.

## 2.2 CONTRASTIVE LEARNING

A common strategy for learning from unlabeled data is *contrastive* learning. In this technique, we seek representations that are *invariant* to minor differences, i.e. they encode a space in which similar objects are closer together, and dissimilar objects are further. A classic strategy for this is to use data *augmentations*, transformations which noticeably alter a datapoint (for example, randomly cropping an image) without changing its essential content (e.g. what the image is of, such as a cat). These transformed versions then become a *positive pair*, while other examples (e.g. an image of a dog) become *negatives*. In audio, contrastive learning has been used extensively to produce high-quality representations for downstream tasks (Al-Tahan & Mohsenzadeh, 2021; Saeed et al., 2021; Ravanelli & Bengio, 2019; Wang & Oord, 2021; Fonseca et al., 2021b). Our approach differs fundamentally from data augmentation strategies commonly used in contrastive learning. Instead of applying transformations to existing audio samples, we generate synthetic audio pairs by perturbing synthesizer parameters, creating positive pairs with causal variations that are difficult to achieve through augmentations. This represents a novel application of synthetic data in the context of general-purpose audio representation learning.

## 2.3 SOUND SYNTHESIS

The toolkit of sound synthesis has evolved to include a variety of hardware and software (Mathews et al., 1969; Pinch & Trocco, 2004; Théberge, 1997). Synthesizers, abstractions of sound synthesis and processing methods often designed to act as musical instruments, are key to this: they expose control parameters that let sound designers guide them to produce desirable sounds for music, film, and many other applications. Accelerated synthesizers (Turian et al., 2021; Cherep & Singh, 2023) have recently allowed much faster-than-realtime sound generation, offering the ability to iteratively tweak parameters to reconstruct sounds (Hagiwara et al., 2022; Shier, 2021) and even match textual descriptions (Cherep et al., 2024). Such approaches highlight the practical utility of synthesizers: lightweight architectures controlled by a limited number of interpretable parameters are capable of producing a diverse array of sounds, often corresponding to well-known categories and concepts (e.g. the sound of waves can often be modeled with time-varying filtered noise). In our work, we leverage SYNTHAX (Cherep & Singh, 2023), to rapidly produce diverse training data with controllable similarity between examples.

## 3 Methods

### 3.1 Data Generation

Our data generation pipeline uses virtual modular synthesizers implemented by SYNTHAX (Cherep & Singh, 2023) in JAX. By default, we use the *Voice* synthesizer architecture (Turian et al., 2021), which can generate perceptually diverse sounds. Our synthesizer consists of several common modules (with parameter-counts in parenthesis):

- Keyboard (2x): Controls the sound's fundamental frequency ($f_0$) and duration.
- Low-Frequency Oscillators (LFOs; 8x each): Two LFOs modulate various aspects of the sound, each with parameters for frequency, modulation depth, initial phase, and amplitude weights across waveforms.
- ADSR Envelopes (5x each): Six envelopes shape the amplitude and modulation signals, each defined by attack, decay, sustain, release, and curvature ($\alpha$).
- Voltage-Controlled Oscillators (VCOs): Includes a sine VCO (3x) with tuning, modulation depth, initial phase, and a square-saw VCO (4x) adding waveform shape.
- Noise Generator: Provides broadband noise without additional parameters.
- Modulations (20x): Weight matrix controlling how modulation sources affect destinations.
- Audio Mixer (3x): Combines outputs of oscillators and noise generator.

In total, the *Voice* synthesizer has 78 parameters. Perturbing these parameters allows us to generate a wide variety of sounds with controlled variations, such as slightly lower or higher pitch, a slightly longer onset or release, or a little more or less noise. In our experiments, we investigate two further architectures: *VoiceFM* has 130 parameters and includes a frequency modulation (FM) operator, and *ParametricSynth* has 2 sine and 2 square-saw oscillators, 1 sine FM and 1 square-saw FM operator, 340 in total. Varying the architecture allows us to investigate whether architectural complexity could affect the quality of representations learned. We generate 1-second sounds by default, for compatibility with most encoders (e.g. VGGish (Hershey et al., 2017)). However, this practice can be extended to longer sounds.

**Synthesis perturbation factor ($\delta$)** A key contribution of our work is synthesizing paired positive samples that sound alike, but are dissimilar due to their synthesis parameters and not only post-hoc effects (e.g. augmentations). This draws on the canonical definition from contrastive learning of positives that are sampled from the same *latent class* (Saunshi et al., 2019).

For a single positive pair, we first sample a parameter vector uniformly randomly $\theta \in [0,1]^{m_S} \sim \mathcal{U}(0,1)$ from the normalized synthesis parameter space, where $m_S$ is the number of control parameters in the given synthesizer. Then, we independently sample two isotropic Gaussian noise vectors $\mathbf{z_1}, \mathbf{z_2} \sim \mathcal{N}(0, \mathbf{I}_{m_S \times m_S})$. We define a parameter $\delta$ that scales this noise, and then produce two perturbed parameter vectors $\theta_i = \theta + \delta \mathbf{z_i} \ \forall_i \in \{1, 2\}$. From these, we clip values back into $[0,1]$ to synthesize two corresponding sounds which serve as positives in the contrastive learning setup.

In principle, $\delta$ controls the distance between the positive pairs and therefore the hardness of the contrastive learning task. Practically, we expect there to be a sweet-spot for $\delta$, considering prior work on mutual information and redundancy in contrastive learning problems (Tian et al., 2020; Tosh et al., 2021) as with very high $\delta$, the parameter vectors may become dominated by noise, resulting in difficulty effectively aligning their representations. Given this, we extensively study the effect of $\delta$ on downstream results.

### 3.2 Real Data

To compare to real audio data, we use sounds from VGGSound (Chen et al., 2020a), a well-known dataset taken from YouTube videos (we only use audio). We use a random sample of 100,000 10-second files and select a random 1-second segment from each file at each iteration to augment. This allows us to fairly compare to our synthetic sounds by keeping duration constant, while still sampling from a variety of real sounds by randomizing the 1-second segments. Though VGGSound has labels included, we do not use them in training these models to keep the self-supervised constraint. Note that VGGSound is currently one of the largest publicly released audio datasets for pretraining, unlike AudioSet (which only releases URLs, not the content itself).

### 3.3 PREPROCESSING, DATA AUGMENTATIONS, AND AUDIO ENCODER

In our experiments, we use VGGish frontend representations (Hershey et al., 2017). We resample audio to 16kHz and obtain mel spectrograms with 64 mel bands and 96 time steps. We use a chain of effects as augmentations (implemented in torch-audiomentations[1]): a high-pass filter (cutoff frequency range 20–800Hz), a low-pass filter (1.2–8kHz), pitch shift (-2 to 2 semitones), time shift (-25% to 25%, rollover enabled), and finally reverberation for which we sample randomly from a set of impulse responses. All augmentations are applied with probability 0.5. We found that this yielded far stronger results than SpecAugment (Park et al., 2019), and so we use this as a comparison point in all our experiments. More details on the augmentation are given in Appendix B. We also test temporal jitter, wherein different 1-second segments are sampled from within the same source clip and treated as positives (Saeed et al., 2021; Spijkervet & Burgoyne, 2021). Our audio encoder is a ResNet18 (He et al., 2016), where we replace the initial layer with a 1-channel convolution to account for the effectively 1-channel spectrogram.

### 3.4 CONTRASTIVE LEARNING

We train for 200 epochs, generating (or sampling) 100,000 sounds per epoch, with a 90%-10% train-validation split. We use a batch size of 768 per GPU with two V100s. The training uses the alignment and uniformity objectives (Wang & Isola, 2020) used in prior work on learning with synthetic data (Baradad Jurjo et al., 2021). We adopt the default parameters for these: $\text{unif}_t = 2$, $\text{align}_\alpha = 2$, and equal weights $\lambda_1 = \lambda_2 = 1$ for both terms. Following this work, we use stochastic gradient descent for optimization, with a maximum learning rate of 0.72 (calculated as $0.12 \times \frac{\text{total batch size}}{256}$) and weight decay $10^{-6}$. The learning rate follows a multi-step schedule with $\gamma = 0.1$, and milestones at 77.5%, 85%, and 92.5% of the total learning epochs. Detailed steps are provided in Algorithm 1. Training with our synthetic data takes approx. 1-2 hours, as the data is generated on the fly in batches, whereas using on-disk datasets with effect chain augmentations can extend training time up to 6-8+ hours.

### 3.5 EVALUATION TASKS

To obtain a broad picture of the quality of our learned representations, we conduct experiments on a range of audio classification tasks from the HEAR (Turian et al., 2022) and ARCH (La Quatra et al., 2024) benchmarks. We use evaluation tasks that focus on general audio understanding, rather than tasks that are highly specialized or domain-specific (e.g., tasks exclusively related to speech or music). Our method aims to learn general-purpose audio representations from synthetic data; therefore, we implement tasks which encompass a broad range of everyday sounds. This aligns with our goal of demonstrating the effectiveness of our representations in diverse real-world scenarios.

These tasks cover a wide range of capabilities including sound classification tasks like ESC-50 (Piczak, 2015), FSD-50k (Fonseca et al., 2021a), and UrbanSound8K (Salamon et al., 2014), vocal affect tasks with and without speech like VIVAE (Holz et al., 2022) and CREMA-D (Cao et al., 2014), musical pitch recognition via NSynth Pitch (5h) (Engel et al., 2017), vocal sound imitation recognition using Vocal Imitations (Kim et al., 2018), and LibriCount (Stöter et al., 2018) for a "cocktail party" style speaker count estimation task. We conduct linear probing experiments using the Adam optimizer for the benchmark-specified epochs with the default learning rate of 0.001 and a batch size of 32.

## 4 RESULTS

### 4.1 BENCHMARK RESULTS

In Table 1, we show results across 8 tasks. The top section features external baselines from the HEAR (Turian et al., 2022) leaderboard and ARCH (La Quatra et al., 2024) benchmark results, first the strongest overall and then only self-supervised. It also includes results from MS-CLAP (Elizalde et al., 2023) linear probing experiments, GURA (Wu et al., 2022) (strongest overall model on HEAR), and finally the original ResNet18 trained on VGGSound (supervised) (Chen et al., 2020a). Note that

---

[1]https://github.com/asteroid-team/torch-audiomentations

---

**Algorithm 1** Our contrastive learning procedure with *audio doppelgängers*. In the training loop, we drop the batch index $i$ for simplicity. We also show the pairwise distance in $\ell_{\text{nuif}}$, though the implementation (via `torch.pdist`) uses a condensed representation.

---

**Require:** Batch size $k$
**Require:** Perturbation factor $\delta$
**Require:** Virtual synthesizer $S$ with $m_S$ parameters
**Require:** Embedding model $M$ with embedding size $m_M$ (512 in our case)
**Require:** Total number of training batches $N_{\text{batches}}$
**Require:** $\ell_{\text{unif}}(\mathbf{X} \in \mathbb{R}^{k \times m_M}) \leftarrow \log\left[\frac{1}{k^2} \sum_{j=1}^{k} \sum_{l=1}^{k} \exp\left(-t\|\mathbf{X}[j] - \mathbf{X}[l]\|_2^2\right)\right]$ where $t = 2$

    **for** $i = 1$ **to** $N_{\text{batches}}$ **do**
       $\boldsymbol{\Theta} \in [0,1]^{k \times m_S} \sim \mathcal{U}(0,1)$                                   {Random batch of parameters}
       $\mathbf{Z}_1, \mathbf{Z}_2 \in \mathbb{R}^{k \times m_S} \sim \mathcal{N}(0, \mathbf{I})$                      {Isotropic Gaussian perturbation noise}
       $\hat{\boldsymbol{\Theta}}_1 \leftarrow \max(0, \min(\boldsymbol{\Theta} + \delta\mathbf{Z}_1, 1))$                      {Clipped perturbed parameters}
       $\hat{\boldsymbol{\Theta}}_2 \leftarrow \max(0, \min(\boldsymbol{\Theta} + \delta\mathbf{Z}_2, 1))$
       $\mathbf{A}_1 \leftarrow S(\hat{\boldsymbol{\Theta}}_1), \mathbf{A}_2 \leftarrow S(\hat{\boldsymbol{\Theta}}_2)$                   {Synthesize audio from parameters}
       $\mathbf{E}_1 \leftarrow M(\mathbf{A_1}), \mathbf{E}_2 \leftarrow M(\mathbf{A_2})$                      {Embedding from model}
       $\mathcal{L}_{align} \leftarrow \frac{1}{k} \sum_{j=1}^{k} \|\mathbf{E}_1[j] - \mathbf{E}_2[j]\|_2^\alpha$ where $\alpha = 2$          {Alignment cost}
       $\mathcal{L}_{uniform} \leftarrow \frac{1}{2}\left[\ell_{\text{unif}}(\mathbf{E}_1) + \ell_{\text{unif}}(\mathbf{E}_2)\right]$                   {Uniformity cost}
       $\mathcal{L}_{total} \leftarrow \lambda_1 \mathcal{L}_{align} + \lambda_2 \mathcal{L}_{uniform}$        {By default, we set $\lambda_1 = \lambda_2 = 1$}
       Update model $M$ using $\mathcal{L}_{total}$
    **end for**

---

HEAR leaderboard results may use MLP probes, whereas ours are linear. We add additional internal baselines, including random weights, synthetic data trained without $\delta$ but with augmentations, and variants of ResNet18 we trained on VGGSound (with augmentations, and alternately with temporal jitter). Finally, we include a selection of our results; the best overall score we achieve using our synthetic approach (first row), followed by the best-performing model trained on data from each of the synthesizer architectures (including *Voice* with augmentations). In Appendix A.3, we provide a full set of results from all model variants: all synthetic datasets for all values of $\delta$, and further baselines less performant than those we present here.

Overall, our best scores uniformly outperform training on VGGSound with augmentations, and outperform training with temporal jitter (the strongest internal baseline) in 6/8 cases. In some cases, these results are also competitive with strong baselines, such as beating the supervised ResNet18 result on 3/8 tasks, CLAP on 2/5, and GURA on 1/6. Additionally, adding further augmentations to our *audio doppelgänger*-based training does not seem to hold significant benefits, despite being highly beneficial when training with synthetic sounds with no $\delta$, suggesting the $\delta$-based perturbations are already sufficiently strong. All this is accomplished without these models seeing any real sounds during pretraining. Finally, *Voice* with $\delta = 0.25$ is the strongest synthetic-trained model overall, being the top performer on 5/8 tasks, but we note that there is some inter-task variability in the best synthesizer and delta.

## 4.2 CHARACTERIZING THE DATA DISTRIBUTION

Here, we focus on understanding the distribution of synthetic sounds and how they differ from natural sound properties. We primarily use our alternate training set, VGGSound (Chen et al., 2020a), for these measures. Unless specified otherwise, we use a randomly sampled (for VGGSound) or generated (for synthetic) set of 1000 sounds for each given dataset used for these characterizations. Our goal is to help understand what properties of the synthetic data make it useful for representation learning, given its strong performance.

| Data/Model | ESC | US8K | VIV | NSyn | C-D | FSD | VI | LCount |
|---|---|---|---|---|---|---|---|---|
| **External Baselines** | | | | | | | | |
| HEAR/ARCH Top | 96.65 | 79.09 | 44.28 | 87.80 | 75.21 | 65.48 | 22.69 | 78.53 |
| HEAR/ARCH SSL | 80.50 | 79.09 | 44.28 | 52.40 | 75.21 | 50.88 | 18.48 | 78.53 |
| MS-CLAP Linear | 89.95 | 82.29 | – | – | 23.15 | 50.24 | – | 54.51 |
| GURA (HEAR) | 74.35 | – | – | 38.20 | 75.21 | 41.32 | 18.48 | 68.34 |
| VGGSound Sup. | 87.45 | 77.57 | 39.38 | 43.80 | 54.36 | 43.76 | 14.06 | 56.10 |
| **Internal Baselines** | | | | | | | | |
| Random Init. | 22.45 | 55.03 | 33.81 | 36.20 | 38.91 | 9.03 | 2.43 | 44.91 |
| *Voice* (Ours, No-$\delta$, Aug.) | 48.65 | 59.46 | 36.31 | 32.80 | 46.32 | 16.88 | 7.12 | 47.64 |
| VGGSound SSL (Aug.) | 48.85 | 61.91 | 32.67 | 39.60 | 47.86 | 19.63 | 6.03 | 53.46 |
| VGGSound SSL (Jitter) | 52.95 | 63.82 | 38.12 | 14.20 | **50.03** | 24.02 | 3.43 | **69.77** |
| **Audio Doppelgängers (Ours)** | | | | | | | | |
| Best Synthetic | **58.90** | **66.71** | **39.45** | **44.40** | 48.43 | **24.12** | **9.15** | **58.60** |
| *Voice* ($\delta = 0.25$) | **58.90** | **66.71** | **39.45** | 32.20 | 48.24 | **24.12** | **9.15** | 52.95 |
| *Voice* ($\delta = 0.25$, Aug.) | 58.75 | 65.01 | 34.81 | **44.40** | 46.17 | 21.76 | 8.54 | 50.70 |
| *VoiceFM* ($\delta = 0.25$) | 57.20 | 65.11 | 38.48 | 35.20 | 48.43 | 22.15 | 6.96 | 54.00 |
| *Parametric* ($\delta = 0.25$) | 50.55 | 62.83 | 37.91 | 37.60 | 46.77 | 18.68 | 5.70 | 54.72 |

Table 1: Evaluation results on a suite of tasks including (from left to right) ESC-50 (Piczak, 2015), UrbanSound8k (Salamon et al., 2014), VIVAE (Holz et al., 2022), NSynth Pitch 5h (Engel et al., 2017), CREMA-D (Cao et al., 2014), FSD50k (Fonseca et al., 2021a), Vocal Imitation (Kim et al., 2018), and LibriCount (Stöter et al., 2018). For internal baselines, we only bold tasks where the baseline beats the best synthetically trained result. Results for all synthetic variants are in Appendix A.3.

### 4.2.1 EMBEDDING SIMILARITY

First, we look at the distribution of synthetic sound pairs and establish that $\delta$ meaningfully controls (a proxy for) perceptual or semantic dissimilarity. We operationalize this using LAION-CLAP embeddings (Wu et al., 2023), since they are trained on a large variety of sounds with semantic descriptions associated. Figure 2A shows how the average cosine similarity decreases monotonically with increasing $\delta$ for all 3 synthesizers. Figure 2B provides a view of how $\delta$ affects the geometry of the embedding space. Here, we plot the first two principal components of the CLAP embeddings along with the path length for each positive pair of synthesized samples from a *Voice* synthesizer. As $\delta$ increases, the path lengths increase and overlap more resulting in less clear separation of positive pairs from negatives. We view this as a signal that we can effectively control the hardness of the contrastive task using $\delta$, the perturbation factor.

### 4.2.2 SIMILARITIES AND DIFFERENCES FROM REAL DATA

Next, we compare the synthetic data distribution to VGGSound (Chen et al., 2020a) data. Figure 3A compares a selection of features' distributions between several dataset variants. For synthetic datasets, we have *Voice*, *VoiceFM*, and *ParametricSynth* variants. For real datasets, we have VGGSound. We first compare to randomly sampled 1-second chunks. Here, the synthetic sounds match several feature distributions well, such as Inharmonicity (Peeters, 2004), Odd-to-Even Harmonic Ratio (Martin & Kim, 1998), Pitch Salience (Ricard, 2004), and, to a lesser extent, Spectral Flatness (Peeters, 2004). However, the synthetic sounds have higher Spectral Flux (Tzanetakis & Cook, 1999) and Complexity (Laurier et al., 2010). Note that *ParametricSynth* also has lower pitch salience. We believe this is due to its larger mixture of sound generators which reduce salience of particular pitches.

Based on these results, we hypothesize that one potential reason the synthetic sounds could be useful for training is the informativeness of the samples. The larger amount of spectral change and higher complexity in terms of peaks could expose the model to more different kinds of sounds rapidly. To try to match these attributes, we produce mixtures of VGGSound, since mixed sounds may have more peaks and variation than individual samples. In VGGSound-Mix 5s, we take 5 arbitrary seconds from each sound and layer them into a 1-second sample. In VGGSound-Mix 10s, we do the same with all 10 seconds available. We show (Figure 3) that these get closer to the synthetic distributions on these

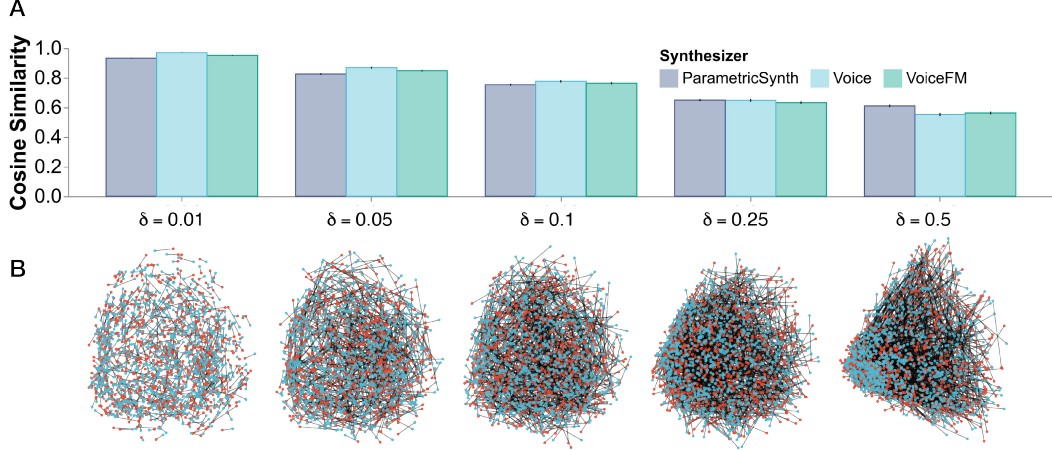

Figure 2: **(A: Top)** Average CLAP (Wu et al., 2023) embedding cosine similarity between positive pairs for different architectures and different values of δ. **(B: Bottom)** PCA of CLAP embeddings for sounds generated with the *Voice* architecture, with line segments showing distances between paired examples. Red and blue points are paired positive instances. Across both plots, as δ increases, the positive pairs systematically become more perceptually dissimilar (via the CLAP embedding proxy).

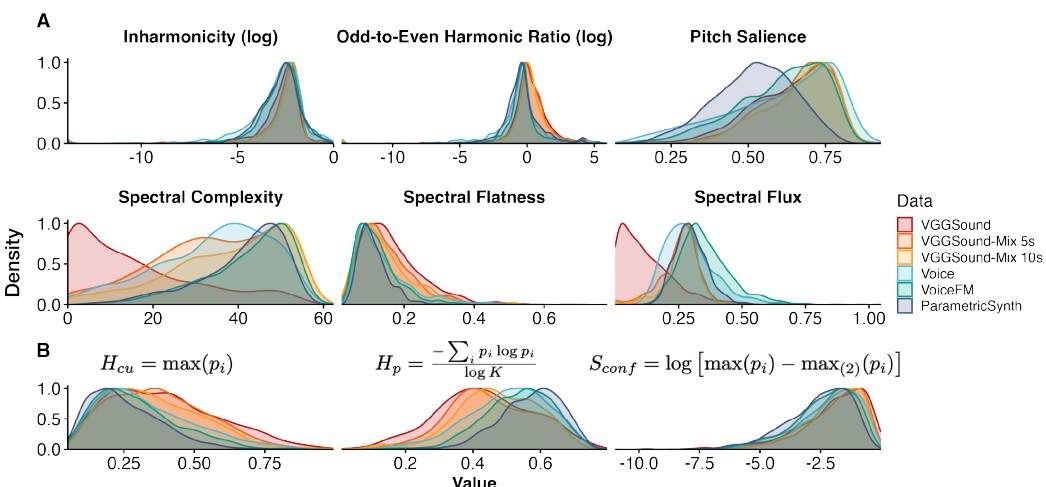

Figure 3: Comparisons of synthetic and real sound data (VGGSound (Chen et al., 2020a)) on **(A: Top)** spectral features and **(B: Bottom)** causal uncertainty. Spectral features of synthetic sounds partially replicate real sounds, but exhibit differences in complexity and flux. Synthetic sounds are also more causally ambiguous, indicating a distribution shift. Using dense mixtures of real sounds partially closes these gaps, suggesting the synthetic sounds are different in part due to their density of auditory information.

features, without deviating on other features. These data distributions allow us to assess whether the benefits of synthetic data are largely driven by the change and informativeness of the signals. In Appendix A.3, we present results from models trained on these mixture distributions, and obtained mixed results, suggesting other factors of the synthesized sounds may also be important beyond this.

### 4.2.3 CAUSAL UNCERTAINTY

We also consider causal uncertainty (Ballas & Sliwinski, 1986; Ananthabhotla et al., 2019; Boger et al., 2021), a factor that we intuitively expect to be different for the synthetic sounds. Helmholtz famously discussed perception in terms of unconscious causal inference from sensory input (Helmholtz, 1867), but the synthetic sounds have no physical causes and do not come from well-understood categories. In Figure 3B, we plot 3 proxies for causal uncertainty derived from probabilities of an AST classifier (Gong et al., 2021) trained on AudioSet. We use the formulation from prior work of $H_{cu}$, the maximum predicted probability (Ananthabhotla et al., 2019; Boger et al., 2021). We also propose two simple metrics to corroborate this: $H_p$ the (normalized) entropy of the output probability distribution, and a confidence score $S_{conf}$, the difference in probability between the most and second-most probable classes (log-scaled for the plot). Across all, the synthetic sounds are more causally uncertain than the real sounds. However, as with the spectral feature distributions, using mixtures of VGGSound (Chen et al., 2020a) clips moves the real distribution slightly closer to the synthetic distribution per $H_{cu}$ and $H_p$. We speculate that exposure to more causally uncertain sounds might be subtly helpful for representation learning; for example, they may contain diverse features that aid generalization to more ambiguous sounds present in downstream tasks. We characterize this as another important distributional difference between the synthetic sounds and realistic sounds from datasets such as VGGSound.

### 4.2.4 SIMILARITY TO TARGET DISTRIBUTIONS

Another lens we can use to understand the effectiveness of training on synthetic data is in terms of the distribution of sounds in the target downstream tasks. A common metric to compare sound distributions is the Fréchet Audio Distance (FAD) (Kilgour et al., 2018). For simplicity, we use the canonical formulation based on VGGish embeddings, though there are some limitations of this (Gui et al., 2024; Tailleur et al., 2024), and we use either the validation sets or first multi-fold splits of the target task audio. Table 2 shows that for ESC-50 (Piczak, 2015), VGGSound is closer in distribution to the target sounds, likely due to ESC-50's focus on environmental sounds. For all other tasks, however, the synthetic sounds achieve a lower FAD, suggesting they may better capture task-relevant features for these tasks' sounds. This finding echoes a study of MMDs in torchsynth (Turian et al., 2021), where the *Voice* architecture shows higher-than-expected similarity to FSD50k sounds. We hypothesize that the synthetic training allows the model to see a wide variety of spectral behavior rapidly, in a way that supports an array of tasks.

| Dataset | ESC-50 | FSD50k | LibriCount | NSynth | CREMA-D | Vocal Imitation |
|---|---|---|---|---|---|---|
| *Voice* | 17.39 | **13.37** | **16.67** | **12.83** | **18.55** | **11.64** |
| *VoiceFM* | 18.48 | 15.91 | 17.67 | 14.49 | 21.24 | 13.66 |
| *ParametricSynth* | 18.75 | 19.44 | 21.04 | 17.42 | 25.33 | 17.32 |
| VGGSound | **6.71** | 25.33 | 29.09 | 27.67 | 33.83 | 27.75 |
| VGGSound-Mix 5s | 8.81 | 26.17 | 30.02 | 28.70 | 34.35 | 29.05 |
| VGGSound-Mix 10s | 9.30 | 26.09 | 30.15 | 28.88 | 34.06 | 29.16 |

Table 2: FAD (Kilgour et al., 2018) scores between different synthetic/real datasets and target downstream task data distributions, computed using either validation sets or the first fold (for multi-fold datasets). For 5/6 tasks, *Voice* achieves the lowest FAD despite containing synthetic sounds. On ESC-50, however, the VGGSound distribution appears to be closest.

### 4.3 ABLATIONS AND SENSITIVITY ANALYSIS

In Figure 4, we study the effect of the perturbation factor $\delta$ on downstream task performance across all tasks for the strongest (*Voice*) architecture with and without additional (FX chain based) augmentations. Overall, we found in early experiments that $\delta = 0.25$ gives the best results. This is observed on the full suite of tasks (with notable exceptions like NSynth without augmentations, and LibriCount overall). In Appendix A.2, we discuss why this value might be strongest from the perspective of alignment and uniformity results.

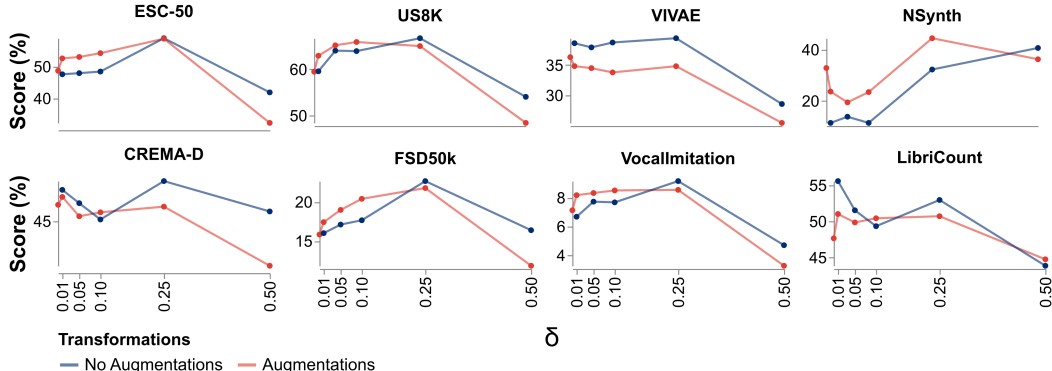

Figure 4: Scores with the *Voice* architecture and different values of $\delta$ for evaluation tasks in Table 1 with and without augmentations. $\delta = 0.25$ tends to give the best results overall.

## 5 LIMITATIONS

Our study demonstrates the efficacy of our approach using established architectures like ResNets, balancing computational efficiency with the goal of producing generalizable results. While we focused on these architectures, our findings lay the groundwork for future investigations with larger encoders such as AST (Gong et al., 2021); this is a straightforward extension. Our research also focused on a clear comparison between synthetic and real data, which allowed us to rigorously evaluate our method's effectiveness. The potential for hybrid approaches, combining synthetic and real data, has a wide range of possibilities in mixing strategies and fine-tuning techniques. These can all be explored without changing our method itself.

The isometric Gaussian noise perturbation proved highly effective, despite its simplicity, since it changes identifiable attributes (e.g. pitch) subtly. This success points to the robustness of our method, while also highlighting opportunities for more sophisticated perturbation strategies to further enhance it. Future work could explore anisotropic perturbations that account for parameter relationships. Adaptive or learned perturbation strategies could also offer significant advancements. Additionally, our evaluation centered on widely used classification benchmarks to create a foundational assessment of the method's performance. Expanding this evaluation to include metrics such as representation disentanglement could offer additional insights into the quality and utility of the synthetic data.

On a broader note, we believe it's important to examine synthetic data-generating procedures for possible biases, similar to the scrutiny applied to real datasets. Though we think this procedure can mitigate some of the biases in real datasets, different synthesizer architectures, values of $\delta$, and other decisions might inadvertently produce performance gaps for different tasks, applications, and downstream populations of use. We evaluated on a wide range of tasks in part to explore this possibility, but further evaluations would be helpful to assess these impacts.

## 6 CONCLUSION

Further improvements in auditory understanding depend greatly on the data underlying new models. In this work, we examined the value of synthetic data for learning representations of sound. We presented a method that perturbs the parameters of random synthetic sounds to generate *audio doppelgängers*, distinct yet similar sounds that provide a strong signal for contrastive learning. Through a comprehensive set of experiments, we showed how this approach can yield strong results on a wide range of tasks. We will release our code and models to enable the community to experiment with synthetic data sources for audio understanding, and hope this approach will help expand the machine learning toolkit for audio processing.

ACKNOWLEDGEMENTS

Manuel received the support of a fellowship from "la Caixa" Foundation (ID 100010434). The fellowship code is LCF/BQ/EU23/12010079. The authors acknowledge the MIT SuperCloud and Lincoln Laboratory Supercomputing Center for providing resources that have contributed to the research results reported within this paper. We extend our heartfelt thanks to all participants in the listening study.

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

## A    ADDITIONAL RESULTS

### A.1    COMPARISON OF DIFFERENT ARCHITECTURES ACROSS TASKS

In Figure 5 we show the relative performance of models trained with data from different synthesizer architectures with $\delta = 0.25$. These results illustrate that, though *Voice*-generated sounds appear strongest overall, there is some task specialization of these different synthesis approaches. For example, on LibriCount and NSynth, *Voice* is the lowest performer here.

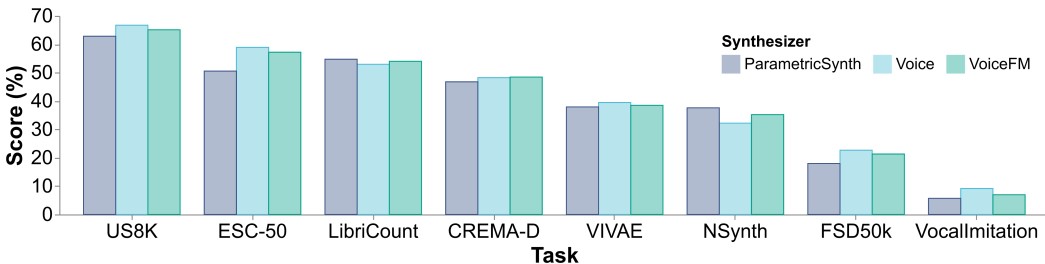

Figure 5: Scores with a fixed $\delta = 0.25$ and different synthesizer architectures for a suite of tasks including (from left to right) UrbanSound8k (Salamon et al., 2014), ESC-50 (Piczak, 2015), Libri-Count (Stöter et al., 2018), CREMA-D (Cao et al., 2014), VIVAE (Holz et al., 2022), NSynth Pitch 5h (Engel et al., 2017), FSD50k (Fonseca et al., 2021a), and Vocal Imitation (Kim et al., 2018)

### A.2    EFFECTS OF INCREASING PERTURBATION FACTOR $\delta$ ON TRAINING

We seek to understand how increasing $\delta$ impacts the training dynamics. In particular, the alignment and uniformity objectives are in tension (Wang & Isola, 2020). A small $\delta$ leads to easy positive pairs (high similarity), resulting in low alignment cost but potentially poor generalization. Conversely, too large $\delta$ produces hard positive pairs (low similarity), increasing the alignment cost but potentially hindering optimization. The optimal $\delta$ should balance this trade-off, however the complexity of the synthesizer function and the embedding function make deriving a closed-form solution for this infeasible. As such, we must explore the effect empirically.

Figure 6 shows the impact of different $\delta$ on the final validation value of the alignment and uniformity costs respectively. Alignment cost increases monotonically with $\delta$, which shows the increased difficulty of aligning increasingly distant pairs. Uniformity has an inverted-U-shaped relationship with $\delta$, suggesting that as the model struggles to align positives with moderate noise driven variation, it incurs a cost in uniformity in order to do so (e.g. creating clusters). With large $\delta$, the amount of noise present is significant, alignment is difficult, and the representations can be more spread out. The theoretically optimal value of $L_{unif}$ is $-2t = 4$, which all values of $\delta$ remain close to. In Figure

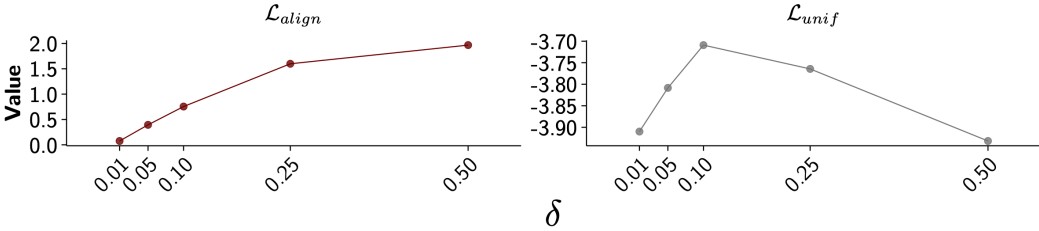

Figure 6: Final validation scores showing the effect of $\delta$ on $\mathcal{L}_{align}$ and $\mathcal{L}_{unif}$. $\mathcal{L}_{align}$ increases monotonically with $\delta$, since the difficulty of aligning more distinct samples goes up. $\mathcal{L}_{unif}$, on the other hand, shows an inverse-U-shaped relationship with $\delta$.

8 of Wang & Isola (2020), the best performance with a more complex task and encoder (review classification) is observed when alignment is on the higher side (but not the maximum), and uniform is low (close to the optimal value). In our experiments, $\delta = 0.25$ gets closest to this, and we observe it to be the strongest as well.

## A.3 RESULTS FOR ALL VARIANTS

We give results for all synthetic model variants below, in Table 3.

| Data/Model | ESC | US8K | VIV | NSyn | C-D | FSD | VI | LCount |
|---|---|---|---|---|---|---|---|---|
| **External Baselines** | | | | | | | | |
| HEAR/ARCH Top | 96.65 | 79.09 | 44.28 | 87.80 | 75.21 | 65.48 | 22.69 | 78.53 |
| HEAR/ARCH SSL | 80.50 | 79.09 | 44.28 | 52.40 | 75.21 | 50.88 | 18.48 | 78.53 |
| MS-CLAP Linear | 89.95 | 82.29 | – | – | 23.15 | 50.24 | – | 54.51 |
| GURA (HEAR) | 74.35 | | – | 38.20 | 75.21 | 41.32 | 18.48 | 68.34 |
| VGGSound Sup. | 87.45 | 77.57 | 39.38 | 43.80 | 54.36 | 43.76 | 14.06 | 56.10 |
| **Internal Baselines** | | | | | | | | |
| Random Init. | 22.45 | 55.03 | 33.81 | 36.20 | 38.91 | 9.03 | 2.43 | 44.91 |
| *Voice (Ours, No-$\delta$, Aug.)* | 48.65 | 59.46 | 36.31 | 32.80 | 46.32 | 16.88 | 7.12 | 47.64 |
| VGGSound SSL (Aug.) | 48.85 | 61.91 | 32.67 | 39.60 | 47.86 | 19.63 | 6.03 | 53.46 |
| VGGSound SSL (Jitter) | 52.95 | 63.82 | 38.12 | 14.20 | **50.03** | 24.02 | 3.43 | **69.77** |
| VGGSound-Mix 5s | 43.95 | 59.69 | 33.31 | 40.80 | 46.10 | 14.71 | 5.95 | 52.57 |
| VGGSound-Mix 10s | 42.95 | 57.40 | 32.03 | 40.20 | 46.57 | 15.77 | 6.43 | 51.07 |
| **Audio Doppelgängers (Ours)** | | | | | | | | |
| Best Synthetic | **58.90** | **66.71** | **39.45** | **44.40** | **48.43** | **24.12** | **9.15** | **58.60** |
| *Voice ($\delta = 0.01$)* | 47.55 | 59.56 | 38.62 | 11.40 | 47.53 | 17.15 | 6.67 | 55.56 |
| *Voice ($\delta = 0.05$)* | 47.90 | 64.02 | 37.93 | 13.80 | 46.45 | 17.77 | 7.72 | 51.52 |
| *Voice ($\delta = 0.10$)* | 48.40 | 63.92 | 38.74 | 11.40 | 45.13 | 18.40 | 7.67 | 49.32 |
| *Voice ($\delta = 0.25$)* | **58.90** | **66.71** | **39.45** | 32.20 | 48.24 | **24.12** | **9.15** | 52.95 |
| *Voice ($\delta = 0.50$)* | 41.85 | 54.03 | 28.54 | 40.60 | 45.78 | 17.14 | 4.69 | 43.85 |
| *VoiceFM ($\delta = 0.01$)* | 42.40 | 59.89 | 36.58 | 9.20 | 44.31 | 15.34 | 5.15 | 57.13 |
| *VoiceFM ($\delta = 0.05$)* | 42.90 | 62.96 | 36.54 | 14.20 | 44.93 | 15.64 | 5.79 | 50.61 |
| *VoiceFM ($\delta = 0.10$)* | 44.80 | 62.03 | 35.73 | 14.80 | 43.99 | 15.67 | 5.60 | 50.56 |
| *VoiceFM ($\delta = 0.25$)* | 57.20 | 65.11 | 38.48 | 35.20 | 48.43 | 22.15 | 6.96 | 54.00 |
| *VoiceFM ($\delta = 0.50$)* | 43.50 | 60.98 | 39.04 | 12.20 | 44.17 | 15.25 | 6.06 | 51.07 |
| *Parametric ($\delta = 0.01$)* | 39.50 | 58.95 | 36.87 | 12.20 | 42.16 | 13.92 | 4.53 | **58.60** |
| *Parametric ($\delta = 0.05$)* | 40.15 | 57.22 | 35.11 | 14.60 | 42.65 | 12.87 | 4.78 | 55.37 |
| *Parametric ($\delta = 0.10$)* | 42.50 | 59.65 | 34.12 | 14.20 | 43.01 | 13.41 | 4.97 | 53.43 |
| *Parametric ($\delta = 0.25$)* | 50.55 | 62.83 | 37.91 | 37.60 | 46.77 | 18.68 | 5.70 | 54.72 |
| *Parametric ($\delta = 0.50$)* | 41.15 | 56.86 | 35.41 | 10.40 | 41.73 | 12.76 | 4.48 | 54.27 |
| *Voice ($\delta = 0.01$, Aug.)* | 52.55 | 62.92 | 34.82 | 23.60 | 46.96 | 18.18 | 8.17 | 51.01 |
| *Voice ($\delta = 0.05$, Aug.)* | 53.00 | 65.17 | 34.49 | 19.40 | 45.39 | 19.79 | 8.32 | 49.84 |
| *Voice ($\delta = 0.10$, Aug.)* | 54.20 | 65.89 | 33.78 | 23.40 | 45.71 | 20.38 | 8.50 | 50.42 |
| *Voice ($\delta = 0.25$, Aug.)* | 58.75 | 65.01 | 34.81 | **44.40** | 46.17 | 21.76 | 8.54 | 50.70 |
| *Voice ($\delta = 0.50$, Aug.)* | 32.25 | 48.40 | 25.41 | 36.20 | 41.38 | 11.82 | 3.26 | 44.74 |

Table 3: Complete results for all model variants.

# B ADDITIONAL DETAILS ON TRAINING

## B.1 AUGMENTATION BATCHING

Due to practical considerations in batching and memory management, augmentations are applied differently for real and synthetic data. In real data, augmentations are applied per-example within distributed data-loading workers. Synthetic data is batch-generated within the main process to avoid concurrency issues between JAX's multithreading and PyTorch's data loading. Individually augmenting examples in this synthetic data environment is prohibitively slow. As a solution, we mini-batched augmentations with a default size $\leq 100$. This allows us to memory-efficiently leverage GPU

processing and introduces variation within each training batch. While per-example augmentations might further enhance performance of synthetic data with augmentations, we believe our current approach is a conservative yet effective option and expect minimal impact.

