# OpenReview forum: "Contrastive Learning from Synthetic Audio Doppelgängers"
_ICLR.cc/2025/Conference — ICLR 2025 Poster_

### Official Review · Reviewer_gLEZ · 2024-11-01

**Soundness:** 4
**Presentation:** 3
**Contribution:** 3
**Rating:** 8
**Confidence:** 4

**Summary:**

This paper introduces a novel approach to contrastive learning for audio representations using synthetic audio doppelgängers. By generating paired synthetic audio samples with controlled variations through sound synthesizer parameter perturbations, the authors address the limitations of traditional data augmentation techniques. This method surpasses results from large-scale audio datasets in several tasks, offering a lightweight, hyperparameter-efficient solution that requires no data storage. The synthetic data approach provides a rich source of contrastive information that enhances the model's ability to learn robust audio representations. The paper's contributions are well-supported by comprehensive experiments across multiple audio classification benchmarks.

**Strengths:**

The synthetic audio doppelgänger concept introduces a unique and controlled approach to creating positive pairs for contrastive learning, which has not been explored in this way for audio data. Comprehensive experiments across multiple benchmarks substantiate the effectiveness of the approach, showing competitive performance with or even surpassing real-data baselines. This method reduces the need for large, labeled datasets, offering a scalable, lightweight solution suitable for a variety of audio applications.

**Weaknesses:**

While the simplicity of using a single hyperparameter δ is a strength, it could also be a limitation. The performance is sensitive to the choice of delta, and further analysis could be helpful in understanding how to optimize this parameter across different tasks. The paper mentions the potential for hybrid approaches combining synthetic and real data but does not explore this in detail. A comparison with hybrid training strategies could provide more insights into the benefits of the proposed method.

**Questions:**

1. How does the choice of synthesizer architecture affect the quality of the learned representations? Is there a significant difference in performance across different synthesizers beyond the three tested?
2. Could the approach be extended to longer audio sequences, and what impact would this have on the quality of representations and training time?
3. Is there potential for further improving performance by using more complex perturbation strategies, such as adaptive or learned perturbations?
4. How robust is the method to variations in delta across different downstream tasks? Would an adaptive approach to tuning δ during training yield better results?

---

> ### Author Response · Authors · 2024-11-15
>
> Thank you for your review of the paper; we appreciate that you consider our work unique and acknowledge several benefits it offers.
>
> ### Weaknesses
>
> **1.** We agree that one hyperparameter is a strength, but that future work should also explore sampling anisotropic Gaussian perturbations to the parameters for more fine-grained control. However, there is no a priori reason that this should improve the performance since our primary goal is to create minor perturbations to audio signals.
>
> **2.** Similarly, we mention potential hybrid synthetic-real data training and encourage this exploration in future work because there are many possible strategies for combining data sources. Investigating this thoroughly is beyond the scope of this work, but we agree it could potentially unlock even greater performance.
>
> ### Questions
>
> **1.** Since the three architectures we test are quite different, and the performance differences are usually not huge, we believe that most reasonably expressive synthesizer architectures might be similar. However, very restrictive architectures (e.g. one oscillator) would most likely limit performance since the diversity of the generated sounds would be greatly reduced.
>
> **2.** Yes, the synthesizer can easily produce arbitrary duration sounds (though we believe that they may not be as useful at very long sounds due to the ADSR envelope designs). The impact on training time would be minimal, and we suspect the impact on representations learned would be modest as well. This is because we already observe good performance, and our results compare favorably with other existing self-supervised results which were trained with longer audio. We kept the duration fixed to provide a controlled experimental setting, in our case (and 1-second audio is compatible with many preprocessing pipelines, such as that of VGGish).
>
> **3.** Yes, we believe there is potential for this, similar to other curriculum and adversarial learning settings. There are some technical barriers to this (e.g. having a smoothly differentiable, expressive synthesizer), however we agree it would be worth exploring.
>
> **4.** We observed that $\delta=0.25$ works well across tasks with different characteristics. We hypothesize this may be due to an “edge of chaos” type phenomenon, as Zhang et al. [1] observe that the best results come when “the system is structured yet challenging to predict.” As such, we agree an adaptive training protocol may provide some benefits, but also think that finding such an optimal level of structure vs. prediction challenge (through $\delta$) might itself be sufficient.
>
> ### References
>
> [1] Zhang, Shiyang, et al. "Intelligence at the Edge of Chaos." arXiv preprint arXiv:2410.02536 (2024).

---

### Official Review · Reviewer_Y8f6 · 2024-11-01

**Soundness:** 3
**Presentation:** 3
**Contribution:** 2
**Rating:** 6
**Confidence:** 3

**Summary:**

This paper introduces a novel contrastive learning approach for audio representation learning using synthetic data. Instead of augmenting real recordings, the authors leverage a modular synthesizer (SYNTHAX) to generate pairs of synthetic audio "doppelgangers" by randomly perturbing synthesis parameters (controlling timbre, pitch, and envelopes) by a factor \delta. This method creates positive pairs with causally controlled variations, providing rich contrastive information. Training a ResNet18 encoder with alignment and uniformity objectives on these synthetic pairs, the authors demonstrate performance exceeding that of models trained on real data (VGGSound) with augmentations across 8 diverse audio classification benchmarks, while also reducing training time. Analysis reveals the synthetic data exhibits higher spectral complexity and flux, and greater causal uncertainty compared to real audio, potentially contributing to its effectiveness. The hyperparameter \delta, controlling the dissimilarity between doppelganger pairs, is shown to significantly impact performance, with \delta = 0.25 generally yielding optimal results across tasks.

**Strengths:**

1. The paper presents a novel approach to contrastive learning in audio by utilizing entirely synthetic data. This is a significant departure from existing methods that rely on augmenting real recordings and opens up a new avenue for research in self-supervised audio representation learning. The demonstrated performance gains over real data with augmentations highlight the potential impact of this approach.

2. The paper is technically sound, with a clear methodology, appropriate experimental setup, and thorough analysis. The authors carefully consider the impact of the perturbation factor (\delta) and provide insightful comparisons between the synthetic and real data distributions, including analysis of features like spectral complexity and causal uncertainty.

3. The proposed method offers practical benefits, particularly in terms of reduced data requirements and faster training times. By eliminating the need for large-scale real-world audio datasets, the approach can significantly reduce the data burden on practitioners and potentially democratize access to high-quality audio representation learning. This has implications for resource-constrained settings and applications where data privacy is a concern.

**Weaknesses:**

1. The authors claim a self-supervised approach, but their method relies on a pre-trained SYNTHAX model. SYNTHAX itself likely required supervised training with labeled data, making the overall approach dependent on prior supervision. The authors should avoid using terms like "self-supervised" and clearly acknowledge the role of supervised pre-training in their method. The augmentations applied are also inherently limited to the codomain of the pre-trained SYNTHAX model, potentially restricting the diversity of generated sounds.

2. The evaluation focuses solely on detection tasks (classification). While the paper's introduction suggests a more general contribution to audio representation learning,  the lack of estimation or generation tasks in the downstream evaluation limits the scope of the demonstrated benefits.  Including a broader range of downstream tasks would strengthen the claims of general-purpose representation learning.

3. The reliance on a specific synthesizer architecture (SYNTHAX) and the parameterized perturbation method could introduce biases in the synthetic data distribution. While the authors analyze some aspects of this distribution, a more thorough investigation of potential biases and their impact on downstream performance across different audio domains and tasks is warranted. The paper would benefit from a discussion of the limitations of the chosen synthesizer and perturbation method and their potential influence on the generality of the learned representations.

**Questions:**

While the paper demonstrates performance improvements, further investigation into the specific properties of the learned representations (e.g., through probing or visualization) and a comparison with alternative self-supervised methods applied to the same synthetic data would provide a more complete understanding of the contributions of the contrastive framework itself. Is it simply the paired structure and controlled variations that are beneficial, or is the contrastive loss crucial?

---

> ### Author Response · Authors · 2024-11-15
>
> Thank you for the review of the paper; we appreciate that you consider our work novel, technically sound, clear, and thorough.
>
> ### Weaknesses
>
> **1.** This is a misunderstanding. SynthAX is a fast modular synthesizer, and therefore there is no pre-training of any kind—supervised or other—or data required. Instead, it relies on signal processing modules (e.g. oscillators) to produce sound based on the (randomly generated) parameters. Our method is self-supervised and not limited to any existing data diversity.
>
> **2.** Audio benchmarks largely consist of classification and detection tasks, but note that we include NSynth pitch recognition; this is more akin to an estimation task, even though it uses a classification-based formulation.
>
> **3.** As you mentioned, Section 4.2.2 describes similarities and differences between synthetic and real data. We also explore three different synthesizer architectures (Voice, VoiceFM, and ParametricSynth) with different modules and parameters. We agree that future work should explore the impact and biases of synthetic data on downstream tasks, but want to highlight that our randomly generated synthetic data also avoids important biases seen in real data (e.g. class imbalance, quality issues from YouTube-sourced audio, etc.).
>
> ### Questions
>
> Thank you for this suggestion. The range of tasks we evaluate on already offer some insights about what the representations encode (e.g. pitch), but we agree that more probing experiments would be helpful. If you have any dataset suggestions, we will do our best to include these in a revised version of the paper.
>
> We hope this response clarifies and resolves any questions you have. Please let us know if you have additional questions, and we would be glad to address them.

---

> > ### Author Response · Authors · 2024-11-20
> >
> > Let us know if you have any additional questions—or if something remains unclear—and we would be glad to address them. Otherwise, we’d appreciate if you could consider raising your score.

---

> > > ### Comment · Reviewer_Y8f6 · 2024-11-25
> > > **Response**
> > >
> > > I thank the authors for their response! My recommendation will stay as accept

---

### Official Review · Reviewer_kgdM · 2024-11-03

**Soundness:** 3
**Presentation:** 4
**Contribution:** 2
**Rating:** 5
**Confidence:** 4

**Summary:**

This paper proposes to rely exclusively on synthetic audio to learn representations with contrastive learning.
The originality of the approach is that it does not rely on audio augmentations (as in SimCLR) directly applied on the audio to create positive pairs, but on generating pairs of audio using a synthesizer with close parameters.
This method, although simple, manages to be on par with some of the SOTA methods on some tasks HEAR and ARCH.

**Strengths:**

The article is well-written and detailed.
The related work section is quite informative, especially the 2.1§.
Although the idea is pretty simple and straightforward, it is evaluated across multiple tasks, mainly against a more classic baseline trained on VGGSound + augmentations, and show good results, despite being trained solely on synthesized sound from a bespoke synthesizer.

**Weaknesses:**

Only one parameter delta to control the amount of variation for all parameters looks quite restrictive. If this is addressed in the Limitations section, it's still surprising that considering that the $f_0$ and the LFO parameters range are treated the same way.
- Results, if interesting, are not beating existing methods, the analysis is extensive an varied.
- This article may be more appropriate for

**Questions:**

- 1 second -long chunks looks quite short to learn interesting representations
- Why using the alignment and uniformity objective from Wang & Isola over other choices like the one in SimCLR or SigLIP?

---

> ### Author Response · Authors · 2024-11-15
>
> Thank you for the review of the paper; we appreciate that you like the idea and find the paper well-written and detailed.
>
> ### Weaknesses
>
> **1.** We see having one hyperparameter as a strength of our method. The simplest approach for generating positive pairs outperforms training with augmentations on real data. Future work could easily sample anisotropic Gaussian perturbations to the parameters, but there is no a priori reason that this should improve the performance since our primary goal is to create minor perturbations to audio signals.
>
> Intuitively, a small perturbation in parameters leads to a minor difference in the audio (e.g. pitch, loudness, modulation rate) since such parameters were designed for humans to tweak by hand as knobs. Accordingly, we sample perturbations from a Gaussian: often making small tweaks, sometimes larger ones, to obtain a range of perturbations.
>
> **2.** Our results outperform training with augmentations on real data for 6 out of 8 tasks and are only clearly surpassed by models trained via supervised learning, or models with larger architectures, which is expected. However, we are glad you found the analysis to be extensive and varied.
>
> ### Questions
>
> **1.** We agree that 1-second sounds are short, yet we can learn good audio representations, highlighting the potential of this work. Although we train on short sounds, we evaluate our method on datasets with sounds of longer durations and show that it has learned a useful representation. Moreover, we show that our synthetic sounds contain more information per second than realistic sounds, and hypothesize that it might be an advantage. Our method can be easily extended to longer sounds too.
>
> **2.** Wang & Isola's alignment-uniformity objectives [1] offer a more interpretable and efficient alternative to SimCLR or SigLIP by explicitly optimizing two geometric properties: pulling similar samples together while spreading representations uniformly. This provides better control and understanding. Moreover, previous research [2] uses it successfully in the visual domain in a similar setup.
>
> Our method can be easily applied to SimCLR and other such methods as well, without changes.
>
> We hope our responses have resolved your questions and concerns. Please let us know if you have any additional questions and we would be glad to address them as well.
>
> ### References
>
> [1] Wang, Tongzhou, and Phillip Isola. "Understanding contrastive representation learning through alignment and uniformity on the hypersphere." International conference on machine learning. PMLR, 2020.
>
> [2] Baradad Jurjo, Manel, Jonas Wulff, Tongzhou Wang, Phillip Isola, and Antonio Torralba. "Learning to see by looking at noise." Advances in Neural Information Processing Systems 34 (2021): 2556-2569.

---

> > ### Author Response · Authors · 2024-11-20
> >
> > Let us know if you have any additional questions—or if something remains unclear—and we would be glad to address them. Otherwise, we’d appreciate if you could consider raising your score.

---

### Official Review · Reviewer_1zPK · 2024-11-08

**Soundness:** 2
**Presentation:** 2
**Contribution:** 2
**Rating:** 6
**Confidence:** 4

**Summary:**

The authors propose contrastive learning from synthetic audio doppelgängers using audio synthesizers that can provide rich source of information that other augmentations fail to provide. Additionally they say that the approach is light weight and requires no extra data storage and only has a single hyper-parameter. They propose that this complements existing strategies and is reducing the data burden on practitioners.

**Strengths:**

== The idea appears neat. Ideally all of the real world audio files should reside in a small latent space. We can sample the latent space to get as many audio files as we want. We could use contrastive learning to learn robust audio representations.

== Decent presentation. The author propose a technique present almost all the necessary that is needed for it. They then give benchmarks on standard datasets.

== The algorithm is well explained easy to follow. The paper is a welcome change from LLM and Transformers based approaches that are just going after scale or yet another prompting paper.

**Weaknesses:**

== It took some digging around to figure out how Voice synthesize with 78 parameters able to do audio synthesis of specific categories. If it is using "Creative Text-to-Audio Generation via Synthesizer Programming" the authors should specifically mention it in section 3.1. It defiantly makes the content of the paper self contained and helps understanding it better. This make it poorly reproducible. It would have been great to add the synthesis technique to the paper.

== Contributions mentioned on Page 2 point 2 are not correct and misleading. There are actually several papers that are doing exact same thing -- using synthetic data in some form for doing audio classification. They should be cited. Here are a few. i) Open-Set Tagging Dataset (OST) Mark Cartwright, ii) BLAT Bootstrapping language-audio pertaining based audio set .... iii) Can synthetic audio from generative foundational models assist audio recognition ....iv) Synthetic training set generation using T2A models for environmental sound classification. There exist many more for audio generation tasks which I am not including here but the authors should have included.

== The abstract mentions reducing the data burden. This is not true. We can define several augmentation strategies that can generate augmentations on the fly e.g. in Audiomentations and it is a matter of implementation detail and not about how we store all of the data.

== The main issue I have is with the algorithm itself. The authors mention that delta parameter is tuned and there exists a sweet spot with a high delta corresponding to noise and lower delta make the sound exactly the same with little diversity. Please comment on this. The algorithm itself would only learn to discern in the positive cases similar sounding what are different but from the same category and in the negative cases noisy versions of the same category. However a typical contrastive learning argument e.g .SimCLR also takes into account audio files that are not noisy but yet from a different category. How would the authors consider these kinds of negative examples ? Is this approach better or worse than a typical contrastive learning approach e.g .SimCLR ?

**Questions:**

Please do not carry out extra experiments and make revisions to the paper. Answering these questions would be enough.

== The paper hinges on synthesis perturbation factor delta and makes the assumption that the parameter lie in a straight-line as explained in section 3.1. What is the evidence of this. What if the parameter space theta have a gaussian distribution or any other distribution. What would happen if I perturbed parameter vectors in any other manner. Is this the right way to perturb parameter.

== Why was the paper not compared against different contrastive learning algorithms. How about SimCLR. Would such methods improve ? The author should the taken the Resnet baseline and done the exact same thing with existing augmentations and SimCLR.

== Why are the gold standard numbers so poor ? FSD-50K paper mentions that the setup that does not include data augmentation including well cited Audio Transformers paper on it. So a ResNet-18 should perform well for MAP metric in the internal baseline and not be so off so as to report 9.03 ?

== why having a single parameter a good thing. Why would doing 100s of different augmentation e.g. audiomentation be a bad thing ? It seems like the augmentations chosen to compare against were weak. Why were all the augmentations not carried out ?

== There are several papers solely focussed on audio augmentation. Why were the baselines not compared against audio augmentation papers  ? Why was augmentations such as random mix or others not carried out ?

== The algorithm itself would only learn to discern in the positive cases similar sounding what are different but from the same category and in the negative cases noisy versions of the same category. However a typical contrastive learning algorithm e.g .SimCLR also takes into account audio files that are not noisy but from a different category. How would the authors consider these kinds of negative examples ? Is this approach better or worse than a typical contrastive learning approach e.g .SimCLR ?

== Why would making the distributions similar in Figure 3 be needed? One would rather argue that the model should retain the same category but have the features described in Figure 3 e.g. inharmonicity be as much varied as possible as this can then account for out of distribution categories as well as make the audio representations robust and generalizable.

---

> ### Author Response · Authors · 2024-11-15
>
> Thank you for the review of the paper; we appreciate that you like the idea and find the work refreshing.
>
> ### Weaknesses
>
> **1.** In this paper, we don’t use the method described in “Creative Text-to-Audio Generation via Synthesizer Programming.” We use the same synthesizer (SynthAX), but audio doppelgängers (our method) are generated by randomly sampling parameters instead of by prompting and leveraging already existing audio-language models. Section 3.1 has details regarding the synthesizer and data generation, but we can include more details in the revised version.
>
> **2.** This is a misunderstanding; the papers you mention are not doing the same thing:
>   - OST uses mixtures of real audio, not synthetic sounds. The mixtures are discussed as “synthetic” but the audio source is FSD-50k (real audio).
>   - BLAT uses synthetic captions (Sec. 3.2), not synthetic audio.
>   - The other two recent papers generate audio using generative models trained on real data, thus breaking the fully-synthetic audio assumption we preserve. They also do not use our doppelgangers strategy.
>   - We will clarify all the above in the revised paper.
>
> **3.** This is incorrect. Other methods, even if generating augmentations on the fly, rely on existing data. Our method generates synthetic data itself all the time during training, avoiding the need for existing data (we have no source audio stored on disks at all).
>
> **4.** There are no categories used during training. The sounds are uniformly sampled from synthetic parameters, meaning the positives are perturbed from the same random parameter vector, and the negatives are sampled independently; we do not conditionally sample from any categories. Our same approach could be used with SimCLR, and would work the same way.
>
> ### Questions
>
> **1.** Intuitively, a small perturbation in parameters leads to a minor difference in the audio (e.g. pitch, loudness, etc.). These parameters were designed for humans to tweak by hand as knobs, and so as such, small tweaks correspond to modest changes. Accordingly, we sample perturbations from a Gaussian: often making small tweaks, sometimes larger ones, to obtain a range of perturbations.
>
> **2.** The method works the same way with SimCLR and other contrastive methods. In [1], the authors prove that contrastive objectives (asymptotically) optimize the alignment+uniformity objectives which we use in this work, and show that using alignment+uniformity leads to similar or better downstream performance. This also allows us to better examine the effect of $\delta$ on model performance. As such, we use alignment+uniformity but the method is easily compatible with SimCLR.
>
> **3.** The 9.03% is a random untrained CNN with only linear probe (low performance is expected). The SSL baseline in our paper gets 24% (which our synthetically trained model also gets). The FSD50k baseline you mention is supervised; we reproduce the supervised ResNet18 results as well (i.e. 43%). On the HEAR benchmark, SSL results on FSD50k range from <9% to >60%.
>
> **4.** A single parameter can be more easily specified, but our framework can easily extend up to $N_\theta$ params (for Voice, that’s 78). I.e. the $\delta$  for each parameter can be specified separately for more fine-grained control. This is a large search space, so conducting comprehensive experiments for this within the scope of our paper is impossible. However, we will mention this possibility.
>
> **5.** We tested a comprehensive augmentation suite from audiomentations, as well as temporal Jitter augmentations. Other augmentations we tested (e.g. SpecAugment) got even lower performance than the audiomentations baseline. The search space is large here, so we aimed to give two strong SSL baselines in addition to supervised baselines and external comparisons. Note that our method can also use any such augmentations, in addition to the synthetic doppelgangers based training.
>
> **6.** This is duplicated from Weaknesses #4, please see our previous response.
>
> **7.** If the real distribution is a subset of the synthetic distribution, then we agree it could aid generalization. However, in practice we do not see this; we see misalignment. As such, it is important to explore the feature distributions to check whether this correlates with performance benefits, as we have done.
>
> We hope these comments have resolved your concerns and questions. If you have any additional questions, please let us know and we would be glad to address them.
>
> ### References
>
> [1] Wang, Tongzhou, and Phillip Isola. "Understanding contrastive representation learning through alignment and uniformity on the hypersphere." International conference on machine learning. PMLR, 2020.

---

> > ### Author Response · Authors · 2024-11-20
> >
> > Let us know if you have any additional questions—or if something remains unclear—and we would be glad to address them. Otherwise, we’d appreciate if you could consider raising your score.

---

### Meta-Review · Area_Chair_Ruab · 2024-12-18

**Metareview:**

Thanks for your submission to ICLR.

This paper introduces a novel approach to contrastive learning for audio representations using synthetic audio doppelgängers.  The reviewers were generally positive about this paper going into the rebuttal/discussion, with three of the four reviewers providing mostly positive reviews.  On the plus side, the reviewers noted that the idea was clever, easy to follow, and useful; and that the presentation was good.  On the negative side, there were various questions about the algorithm and evaluation (e.g., the use of a single hyperparameter was noted by multiple reviewers).

Despite prodding from the AC, the reviewers did not engage much with the discussion.  In particular, the most negative reviewer did not address the rebuttal.  I read through the rebuttals and reviews, and felt that they adequately addressed the concerns, particularly of the more negative reviewer.  Further, there was nothing in these concerns that seemed to warrant not accepting the paper.  Indeed, the paper should be of interest to many in the community, particularly those working in audio.

Please keep in mind the reviewer comments when preparing a final version of the manuscript.

**Additional Comments On Reviewer Discussion:**

The reviewers were mostly nonresponsive during the discussion, but the authors did adequately address the reviewer concerns.

---

### Decision · Program_Chairs · 2025-01-22

Accept (Poster)